# Optical Fiber Cladding SPR Sensor Based on Core-Shift Welding Technology

**DOI:** 10.3390/s19051202

**Published:** 2019-03-09

**Authors:** Yong Wei, Jiangxi Hu, Ping Wu, Yudong Su, Chunlan Liu, Shifa Wang, Xiangfei Nie, Lu Liu

**Affiliations:** 1Chongqing Municipal Key Laboratory of Intelligent Information Processing and Control of Institutions of Higher Education, Chongqing Three Gorges University, Chongqing 404100, China; 2College of Electronic & Information Engineering, Chongqing Three Gorges University, Chongqing 404100, China; 20160008@sanxiau.edu.cn (J.H.); 20160040@sanxiau.edu.cn (P.W.); 20160009@sanxiau.edu.cn (Y.S.); 20160010@sanxiau.edu.cn (S.W.); 20160012@sanxiau.edu.cn (X.N.); 3Chongqing Engineering Research Center of Internet of Things and Intelligent Control Technology, Chongqing Three Gorges University, Chongqing 404100, China; guangxianchuangan@njust.edu.cn; 4Department of Physics, Harbin Institute of Technology, Harbin 150001, China; 18b311002@stu.hit.edu.cn

**Keywords:** surface plasmon resonance, fiber cladding SPR

## Abstract

The typical structure of an optical fiber surface plasmon resonance (SPR) sensor, which has been widely investigated, is to produce the SPR phenomenon by the transmission of light in a fiber core. The traditional method is to peel off the fiber cladding by complex methods such as corrosion, polishing, and grinding. In this paper, the transmitted light of a single-mode fiber is injected into three kinds of fiber cladding by core-shift welding technology to obtain the evanescent field directly between the cladding and the air interface and to build the Kretschmann structure by plating with a 50-nm gold film. The SPR sensing phenomenon is realized in three kinds of fiber cladding of a single-mode fiber, a graded-index multimode fiber, and a step-index multimode fiber. For the step-index multimode fiber cladding SPR sensor, all the light field energy is coupled to the cladding, leading to no light field in the fiber core, the deepest resonance valley, and the narrowest full width at half maximum. The single-mode fiber cladding SPR sensor has the highest sensitivity, and the mean sensitivity of the probe reaches 2538 nm/RIU (refractive index unit) after parameter optimization.

## 1. Introduction

Surface plasmon resonance (SPR) technology has become a novel sensing detection method in recent years. The technology is widely applied in the fields of biopharming, environment monitoring, and geological disasters due to its advantages of high sensitivity, rapid response speed, no label, and real-time monitoring [1,2]. Besides the above features of SPR technology, optical fiber SPR sensors have the advantages of a small size of the fiber, anti-electromagnetic interference, anti-corrosion, high temperature resistance, and remote sensing measurement. The optical fiber SPR sensor is the natural extension of SPR sensing technology development and the concrete embodiment to realize the integration, micromotion, high sensitivity, and high reliability of sensors.

According to the principle of SPR detection, the essential condition of SPR sensing is that the evanescent field permeates into the gold film. For the common silica cladding optical fiber, the evanescent field exists in the interface between the fiber core and the cladding because the cladding wraps the fiber core, and the transmission of light between the fiber core and the cladding has total reflection. In order to contact the evanescent field with the gold film, some of the traditional methods to peel off the cladding of the fiber are hydrofluoric acid (HF) corrosion [3,4], polish-grinding on the side face of the fiber [5,6,7,8], and grinding on the end face of the fiber [9,10,11,12]. The fiber core is exposed to the air, and the evanescent field exists in the interface of the fiber core and the air. Then, the surface of the fiber core is plated with a 50-nm gold film and immersed in the detecting solution to build the three-layer structure of the fiber core-50 nm gold film-detecting solution. The evanescent field revealed from the surface of the fiber core enters into the gold film, and SPR occurs. We call this a fiber core-type SPR sensor.

Due to the structure of the optical fiber, for which the cladding completely encloses the fiber core, in order to realize the fiber SPR sensor, another method is that the transmitted light of the fiber core is coupled to the fiber cladding (optical fiber grating [13,14,15,16], tapering [17,18], heterogeneous core structure [19,20]), which leads to the evanescent field directly presenting in the interface between the fiber cladding and the air. In the meantime, the fiber cladding is plated with a 50-nm gold film and immersed in the detecting solution to construct the fiber cladding-50-nm gold film-detecting solution structure. The SPR phenomenon is produced when the evanescent field is revealed from the surface of the optical fiber cladding and enters the gold film. For this kind of sensor, the fiber structure usually cannot be changed to realize the coupling of the energy of the transmission of light from the fiber core to the fiber cladding to obtain a fiber SPR sensor with high efficiency and high stability. We call this a cladding-type SPR sensor.

In this paper, we propose a fiber cladding SPR sensor based on core-shift welding technology. The light is coupled in the fiber cladding and is transmitted a short distance by the core-shift welding technology. When the light is transmitted in the fiber cladding, the evanescent field between the fiber and the cladding is completely exposed to the air to resolve the problem of the hard acquisition of the evanescent field from the fiber-type SPR sensor perfectly. In the meantime, the fiber cladding is plated with a 50-nm gold film. This novel fiber SPR sensor is built based on common fibers using the easiest structure and design. The proposal effectively avoids the complex processing methods, including cladding corrosion, side polishing, grinding on the ends, optical gating, and fused biconical tapering, to obtain the interaction of the evanescent field and the gold film for the fiber SPR sensor, and it increases the repeatability of fiber SPR probe manufacture.

## 2. Experimental Section

In this paper, we propose a fiber cladding SPR sensor based on core-shift welding technology. In order to verify whether the transmission of light is effectively coupled to the three kinds of fiber cladding after core-shift welding of a single-mode fiber, we firstly investigate the optical transmission field of core-shift welding.

### 2.1. The Optical Transmission Field of a Single-Mode Fiber When Using Normalized Welding and Core-Shift Welding

Normalized core welding: The coating layer was peeled off from the single-mode fiber, and the fiber was cut to have a smooth end face by the fiber cutter. The fiber was completely welded face-to-face with another finished single-mode fiber by the fiber fusion splicer. The schematic diagram of the fiber structure after welding is shown as Figure 1a. A 532-nm green light source was injected into the single-mode fiber on the left side, and the end face of single-mode fiber on the right side was cut into a smooth plane. The end face was placed under the objective lens of the microscope to observe the transmission of the optical field from the fiber end face. As shown in Figure 1b, the transmitted light was focused in the 9-μm fiber core. The single-mode fiber on the right side lied flat on the hydrophobic slide, and the eosin solution was added dropwise to observe the output of the optical field from the side face, as shown in Figure 1c. As seen in Figure 1b,c, when the single-mode fiber was normalized welded with another single-mode fiber, the light of the single-mode fiber core on the left was directly injected into the single-mode fiber core on the right side, and the light was transmitted along the fiber core to finally form the emergent light.

Core-shift welding: As shown in Figure 1d, the single-mode fiber was core-shift welded with another single-mode fiber by the fiber fusion splicer. The light of the single-mode fiber core on the left side was coupled to the cladding of the single-mode fiber on the right side, indicated as the red arrow in this figure. The output optical field from the end face of the fiber on the right side is shown in Figure 1e, and the transmitted light completely filled the entire 125-μm fiber cladding. The output optical field from the right side of the fiber face is shown in Figure 1f. According to the above results, the transmitted light of the single-mode fiber core on the left side was effectively coupled to the cladding of the single-mode fiber on the right side, and the evanescent field would be produced in the interface between the right-side single-mode fiber cladding and detecting solution.

### 2.2. Transmission of the Light Field of the Gradient Index Multimode Fiber When Using Normalized Core Welding and Core-Shift Welding

Normalized core welding: As shown in Figure 2a, a single-mode fiber was completely welded face-to-face with the gradient index multimode fiber (GI 50/125-20/250, YOFC, Wuhan, China), characterized by a fiber core of 50 μm in diameter and a numerical aperture of 0.2, by the fiber fusion splicer. The light from the single-mode fiber core on the left side was directly injected into the gradient multimode fiber core on the right side. Due to the characteristic of the self-focusing of the gradient multimode fiber (the refractive index in the middle of the fiber core was higher, and the refractive index was gradually decreased on both sides), the light transmitted and exited according to the red arrow direction of the figure. The output light field from the end face of the gradient refractive index multimode fiber on the right side is shown in Figure 2b. Although the diameter of the transmitted light was slightly enlarged, most of the transmitted light was focused in the middle section of the fiber core. The output light field from the side face of the gradient refractive index multimode fiber on the right side is shown in Figure 2c.

Core-shift welding: As shown in Figure 2d, the single-mode fiber was core-shift welded with the gradient refractive index multimode fiber by the fiber fusion splicer. The light of the single-mode fiber core on the left side was coupled to the gradient refractive index multimode fiber cladding on the right side, shown as the red arrow in the figure. The output light field from the end face of the gradient refractive index multimode fiber on the right side is shown in Figure 2e. The transmitted light filled the entire 125-μm fiber cladding, and meanwhile, the amount of transmitted light in the 50-μm fiber core was less. The output light field from the fiber on the right side is shown in Figure 2f. According to the figures, the transmitted light of the single-mode fiber core on the left side was effectively coupled to the gradient refractive index multimode fiber cladding on the right side, and most of the light power was focused in the fiber cladding.

### 2.3. The Transmission of the Light Field of the Step Refractive Index Multimode Fiber When Using Normalized Core Welding and Core-Shift Welding

Normalized core welding: As shown in Figure 3a, the single-mode fiber was welded with the step refractive index multimode fiber (SI 40/125-22/250, YOFC, Wuhan, China), characterized by a 40-μm diameter fiber core and a numerical aperture (NA) of 0.22, by the fiber fusion splicer. The light from the single-mode fiber core on the left side was directly injected into the step multimode fiber core on the right side, and the light transmitted and exited according to the red arrow indicated in the figure. The output light field from the end face of the step refractive index multimode fiber on the right side is shown in Figure 3b, and the transmitted light completely filled the entire step refractive index multimode fiber core of 40 μm. The output light field from the side face of the step refractive index multimode fiber on the right side is shown in Figure 3c.

Core-shift welding: As shown in Figure 3d, the single-mode fiber was core-shift welded with the step refractive index multimode fiber by the fiber fusion splicer. The light of the single-mode fiber core on the left side was coupled to the step refractive index multimode fiber cladding on the right side, shown as the red arrow in the figure. The output light field from the end face of the gradient refractive index multimode fiber on the right side is shown in Figure 3e. The transmitted light filled the entire fiber cladding of 125 μm, and there was no transmission of light in the 40-μm fiber core. The output light field from the right side face of the fiber is shown in Figure 3f. According to the above results, the transmitted light of the single-mode fiber core on the left side was effectively coupled to the step refractive index multimode fiber cladding on the right side, and all the light power was focused in the fiber cladding.

To solve the problem existing in the two types of proposals for an optical fiber SPR sensor, it is necessary to investigate and realize a novel proposal, which can directly couple the light of the fiber core into all kinds of fiber cladding in an easy and effective way, to carry out the transfer of the evanescent field from the interface between the fiber core and the cladding to the interface between the cladding and the air. Then, a 50-nm gold film was coated on the surface of the fiber cladding to realize SPR sensing in all kinds of fiber cladding. The above proposal is one of the presently emphasized fiber SPR sensor investigations. Namely, it is an urgent issue to construct a fiber SPR sensor, characterized by a new structure, a simple fabrication, a stable property, a low cost, and the application to all kinds of fibers. According to the experimental results for all kinds of fibers that are core-shift welded, we proposed a fiber cladding SPR sensor based on core-shift welding technology: the single-mode fiber was core-shift welded with all common types of fibers in the fiber welding machine. The wide spectrum of the light of the single-mode fiber core was directly coupled to all kinds of sensing fiber cladding to lead to the evanescent field existing at the interface between the light-sensing cladding and the air. The sensing fiber cladding was directly plated with a 50-nm gold film to build the fiber cladding-gold film-sample solution structure. The evanescent field can permeate from the cladding into the nanoscale gold film to produce the SPR phenomenon. This novel proposal can avoid the complex processing technology of peeling off the fiber cladding of the traditional fiber core type SPR sensor and can effectively solve the problems of traditional fiber SPR sensors, such as the difficulty in processing-production and the high cost.

### 2.4. Experimental Setup

The experiment testing platform was built as shown in Figure 4 in order to verify the feasibility and detection sensitivity of the proposed fiber cladding SPR sensor based on the core-shift welding technology. The fiber cladding SPR sensing probe based on the core-shift welding technology was built as follows: firstly, the single-mode fiber was core-shift welded with another single-mode fiber. The large core step refractive index multimode fiber, with a core diameter of 105 μm, was regarded as the light-receiving fiber to fabricate the single-mode fiber cladding SPR sensor as the testing object (the length of the sensing single-mode fiber in the middle was 20 mm and plated with a 50-nm gold film), as shown in Figure 4a. The completed fiber cladding SPR sensing probe was sealed in the reaction tank. The glycerol solution to be tested in the syringe was injected into the reaction tank by a micro-injection pump (LSP01-1A, LongerPump, Baoding, China). The waste liquor after the measurement was dispensed into the waste reservoir, and the refractive index of glycerol solution was calibrated by the Abbe refractive index analyzer (GDA-2S, Gold). An ultra-continuous spectral light source (SuperK compact, NKT Photonics) was injected into the single-mode fiber. The light of the single-mode fiber core was injected into the sensing single-mode fiber cladding by core-shift welding. The sensing single-mode fiber cladding was plated with a 50-nm gold film to produce the SPR phenomenon. The light was then transmitted into the large core step refractive index multimode fiber and finally entered the spectrometer (AQ6373B, Yokogawa, Tokyo, Japan) to collect the attenuation spectrum of SPR.

The schematic diagram in Figure 4a is the single-mode fiber cladding SPR sensor structure based on core-shift welding technology. The single-mode fiber on the left is the light injection fiber in this structure, and the light is injected into the fiber cladding in the middle section of the structure after core-shift welding. The light is transmitted in the middle section of the fiber, which is the SPR sensing fiber of the structure. The large core multimode fiber on the right is the light-receiving fiber of this structure to complete the collection of the light signal.

## 3. Results and Discussions

### 3.1. Results

For the proposed cladding SPR sensor based on the core-shift welding technology, the single-mode fiber was the light injection fiber. The experiment was divided into three groups, namely the core-shift welding single-mode fiber, the gradient refractive index multimode fiber, and the step refractive index multimode fiber, then these fibers were respectively welded with a large core diameter multimode fiber by the normalized welding method to collect light. The fibers after welding were plated with a 50-nm gold film by a plasma sputter (ETD-2000, YLBT, Beijng, China), where a rotary fixture was installed for fixing and rotating the fibers. Finally, we fabricated the single-mode fiber cladding SPR sensing probe, the gradient refractive index multimode fiber cladding SPR sensing probe, and the step refractive index multimode fiber cladding SPR sensing probe. In the constructed experiment testing system, the SPR spectrum of the probes from the three groups is shown in Figure 5 to test the sample solutions successively with a 1.333–1.385 RIU refractive index. In Figure 5a, the single-mode fiber was core-shift welded with another single-mode fiber with a length of 2 cm, and the step refractive index multimode fiber with a core diameter of 105 μm was the light-receiving fiber. In Figure 5b, the single-mode fiber was core-shift welded with gradient refractive index multimode fiber whose core diameter was 50 μm and length was 2 cm, and the step refractive index multimode fiber with the core diameter of 105 μm was the light-receiving fiber. In Figure 5c, the single-mode fiber was core-shift welded with the step refractive index multimode fiber, whose core diameter was 40 μm and length 2 cm, and the step refractive index multimode fiber with the core diameter of 105 μm was the light-receiving fiber.

In Figure 5a, the normalized light intensity of the deepest resonance valley was −0.13. In Figure 5b, the normalized light intensity of the deepest resonance valley was −0.16. In Figure 5c, the normalized light intensity of the deepest resonance valley was −0.22. Namely, the resonance valley became gradually deeper and the SPR effect became gradually stronger when the sensing fibers of the fiber cladding SPR sensors were respectively the single-mode fiber, the gradient refractive index multimode fiber, and the step refractive index multimode fiber. At the bottom left of every test graph, seen from the transmission of the light field corresponding to the different types of sensing fibers when using core-shift welding, the causes of the successive enhancement of the SPR effect were as follows: Light was injected into single-mode fiber cladding, and the optical output power was distributed in the entire fiber. Light was injected into the gradient refractive index multimode fiber cladding, and the optical output power was mainly focused in the cladding with a part of the optical output power in the fiber core. Light was injected in the step refractive index multimode fiber cladding, and the optical output power was completely focused in the cladding. When the sensing fiber was the single-mode fiber, the gradient refractive index multimode fiber, and the step refractive index multimode fiber, the proportion of the optical output power in the cladding increased successively, the evanescent field in the interface between the cladding and the air increased successively, and the resonance valley deepened successively. Namely, the resonance valley was deepest in the step multimode fiber because the entire optical power was in the cladding, producing a stronger evanescent field, more optical power interacts with the gold film, and the SPR phenomenon occurred. A part of the light from the single-mode and the gradient multimode fiber was transmitted in the fiber core or the middle section of the fiber, and the light had no interaction with the gold film, which led to no SPR phenomenon for a part of the optical power.

Secondly, compared with the full width at half maximum in the three types (single-mode fiber, gradient refractive index multimode fiber, and step refractive index multimode fiber) of the fiber SPR curves in Figure 5, the full width at half maximum of the curve had a successive decreasing trend as a whole. Namely, when the sensing fiber was the step refractive index multimode fiber, the full width at half maximum of the curve was narrowest.

Then, in this paper, every SPR curve was normalized again, namely the maximum value and the minimum value of every curve were unified between zero and one. The resonance valley height of every curve was almost aligned, and every curve was further adjusted for easy observation and sensitivity calculation. The renormalization result is shown in Figure 6.

The SPR equation for the sensitivity is S_λn_ = ∂λ_res_/∂n_s_, where S_λn_ is the mean sensitivity, ∂λ_res_ is the shift in the resonance wavelength, and ∂n_s_ is the unit change in the refractive index. When the detecting range of the refractive index was 1.333–1.385 RIU, the single-mode fiber cladding SPR sensor was as shown in Figure 6a with a dynamic range of the resonance wavelength between 655 nm and 787 nm, totaling 135 nm, and a mean sensitivity of 2538 nm/RIU. The sensitivity of the traditional fiber-core-type SPR sensor with the multimode fiber was about 2000 nm/RIU. For the novel fiber cladding SPR sensor we proposed, it had a slightly higher sensitivity than most multimode fiber-core-type SPR sensors; the gradient refractive index multimode fiber is as shown in Figure 6b with a dynamic range of the resonance wavelength between 623 nm and 729 nm, totaling 106 nm, and a mean sensitivity of 2038 nm/RIU; the step refractive index multimode fiber is as shown in Figure 6c with a dynamic range of the resonance wavelength between 622 nm and 732 nm, totaling 110 nm, and a mean sensitivity of 2115 nm/RIU. The single-mode fiber cladding SPR sensor had the highest sensitivity.

### 3.2. Discussion

In this paper, we proposed and fabricated a fiber cladding SPR sensor based on core-shift welding technology. The manufacture parameters that affected the valley depth of the SPR resonance curve, the full width at half maximum, and the sensitivity were being plated with a gold film of a certain thickness, having a certain fiber length of the sensing zone, and a certain offset displacement amount when core-shift welding. The gold film thickness was directly chosen as 50 nm, which is generally accepted; we researched the effect of the sensing fiber length in the middle section and the offset displacement in the core-shift welding step on the fiber SPR sensor properties, respectively. While conducting this research, the single-mode fiber, characterized by the highest sensitivity, was fabricated into the single-mode fiber cladding structure SPR sensor and tested.

#### 3.2.1. Different Sensing Fiber Lengths

When fabricating the structure, the single-mode fiber was core-shift welded with the single-mode fiber with an offset displacement of 36 μm by manually fixing the mode in the fiber fusion splicer. The cutting lengths to control the sensing single-mode fiber in the middle section were respectively 0.5 cm, 1 cm, 2 cm, 3 cm, and 5 cm. Figure 7a–e corresponds to the testing curves, respectively.

Figure 7f shows the normalized light intensity of the deepest resonance valley corresponding to the sensing fiber with different lengths. The result shows that with the gradual increase in sensing fiber length, the resonance valley depth was gradually augmented, and when the sensing fiber length was respectively 0.5 cm, 1 cm, 2 cm, 3 cm, and 5 cm, the deepest resonance valley was −0.09, −0.10, −0.12, −0.18, and −0.42. Secondly, as shown in Figure 7a–e, when the increase of the sensing fiber length reached above 3 cm, the full width at half maximum of the SPR resonance curve obviously widened, and the curve shape was degraded.

Then, the renormalized results of every SPR curve in Figure 7 are shown in Figure 8.

The renormalized experimental data curve is shown in Figure 8, for different fiber lengths of the sensing section. The corresponding relation between the sensing fiber length and sensitivity is shown in Figure 8f; namely, with the increase in sensing fiber length, the sensitivity was gradually decreasing. Taken together, for the proposed fiber cladding SPR sensor based on core-shift welding technology, the effect of sensing fiber length on the sensor properties and the optimized selection parameters was as follows: When the length was less than 1 cm, the sensitivity was increased, but it was difficult to identify due to the too shallow resonance valley and bad resonance curve shape. When the length was above 3 cm, the sensitivity was too low and the resonance valley too wide. The length of 1–3 cm was a suitable choice and was easily recognized due the appropriate sensitivity, resonance valley depth, and full width at half maximum.

#### 3.2.2. Different Displacements of Core-Shift Welding

We investigated the effect of different displacements of core-shift welding on the SPR curve property. When fabricating the structure, the core-shift welding displacement was respectively 12 μm, 24 μm, 36 μm, 42 μm, and 54 μm by manually fixing the mode in the fiber fusion splicer. Under the microscope, sensing fiber length was cut into 2 cm. The testing curves are respectively shown in Figure 9a–e.

Figure 9f shows the normalized light intensity of the deepest resonance valley corresponding to different core-shift welding displacements. The result shows that with the gradual increase in the core-shift welding displacement, the resonance valley depth was first gradually increased and then decreased. Namely, when the core-shift welding displacement was 36 μm, the corresponding resonance valley was the deepest; when the core-shift welding displacement was about half of the difference value between the cladding diameter and fiber core diameter, the resonance valley was the deepest. After calculation, there was no significant effect of the change in the core-shift welding displacement on the sensor sensitivity.

## 4. Conclusions

The optical fiber SPR sensor has a series of advantages such as high sensitivity, remote transmitting of monitored information by docking of the optical network, and rapid response, which has been widely investigated in the field of environmental geological monitoring. Based on the sensing mechanism, the SPR phenomenon is produced depending on the contact between the evanescent field of the transmission of light and the gold field as the prerequisite to realize sensing. However, the fiber evanescent field exists in the interface between the fiber core and the cladding. The traditional method is to peel off the fiber cladding by corrosion, polish-grinding, tapering, and other complex mechanical processing methods and to plate it with a gold film, thus leading to the disadvantage of the difficulty in processing and the bad property repeatability. In this paper, by investigating fiber core-shift welding technology, light is directly injected into the sensing fiber cladding and the evanescent field is obtained at the interface between the cladding and the air to solve the difficulty in obtaining the evanescent field of the SPR sensor creatively. The proposal significantly promotes the technology and structural novelty of the SPR sensor with its high sensitivity.

This paper respectively realizes the single-mode fiber cladding SPR sensor, the gradient refractive index multimode fiber cladding SPR sensor, and the step refractive index multimode fiber cladding SPR sensor. Comparing the sensing fibers, the single-mode fiber, the gradient refractive index multimode fiber, and the step refractive index multimode fiber, the resonance valley of the step refractive index multimode fiber is the deepest because all of the light field energy is entirely coupled to the step refractive index multimode fiber cladding with almost no light field in the fiber core, the strongest evanescent field leaked from cladding, the easiest recognition of the resonance valley, and the most regulated curve. However, the single-mode fiber as the light sensing fiber has the highest sensitivity of 2538 nm/RIU. We further investigated the parameters of the single-mode fiber cladding proposal with the highest sensitivity, the lowest price, and the most easily obtained characteristics. When the length of the SPR sensing fiber in the middle section is 1–3 cm, we have a suitable sensitivity, a good resonance valley depth index, an appropriate full width at half maximum, and easy recognition. The core-shift welding displacement of 36 μm has the deepest resonance valley and the best effect.

## Figures and Tables

**Figure 1 sensors-19-01202-f001:**
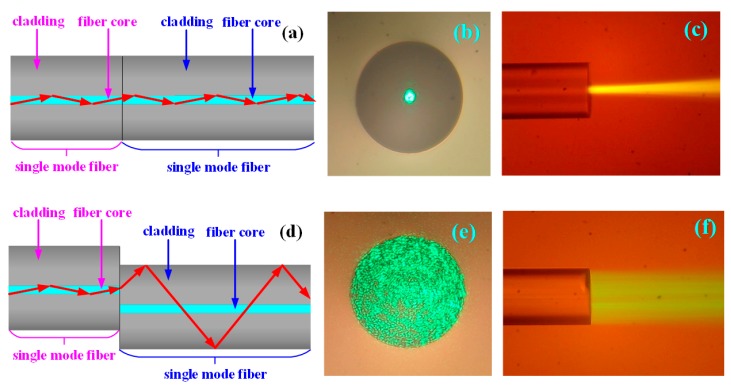
The investigation of the transmission of the light field of the single-mode fiber when normalized core welding and core-shift welding are used. (**a**) The schematic diagram of the single-mode fiber structure and light transmission when using normalized core welding. (**b**) Microphotograph of the light field from the end face of the single-mode fiber when normalized core welding is used. (**c**) Microphotograph of the light field from the side face of the single-mode fiber when using normalized core welding. (**d**) The schematic diagram of the single-mode fiber structure and light transmission when core-shift welding is used. (**e**) Microphotograph of the light field from the end face of the single-mode fiber when core-shift welding is used. (**f**) Microphotograph of the light field from the side face of the single-mode fiber when using core-shift welding.

**Figure 2 sensors-19-01202-f002:**
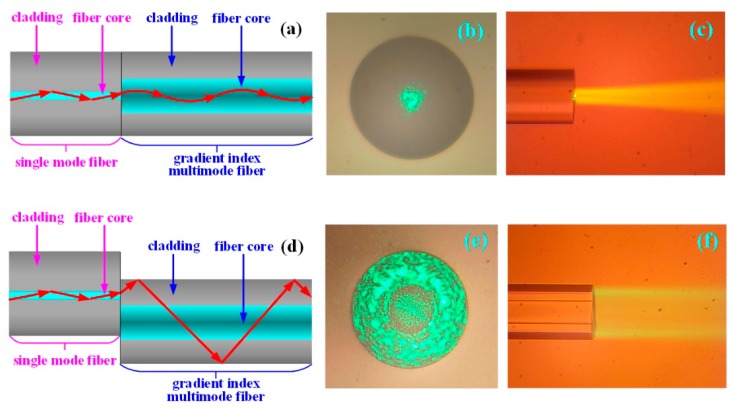
The investigation of the transmission of the light filed of the gradient refractive index multimode fiber when normalized core welding and core-shift welding are used. (**a**) The schematic diagram of the structure and the light transmission of the gradient refractive index multimode fiber when normalized welding is used. (**b**) Microphotograph of the light field from the end face of the gradient refractive index multimode fiber when normalized core welding is used. (**c**) Microphotograph of the light field from the side face of the gradient refractive index multimode fiber when normalized core welding is used. (**d**) The schematic diagram of the gradient refractive index multimode fiber structure and light transmission when core-shift welding is used. (**e**) Microphotograph of the light field from the end face of the gradient refractive index multimode fiber when core-shift welding is used. (**f**) Microphotograph of the light field from the side face of the gradient refractive index multimode fiber when core-shift welding is used.

**Figure 3 sensors-19-01202-f003:**
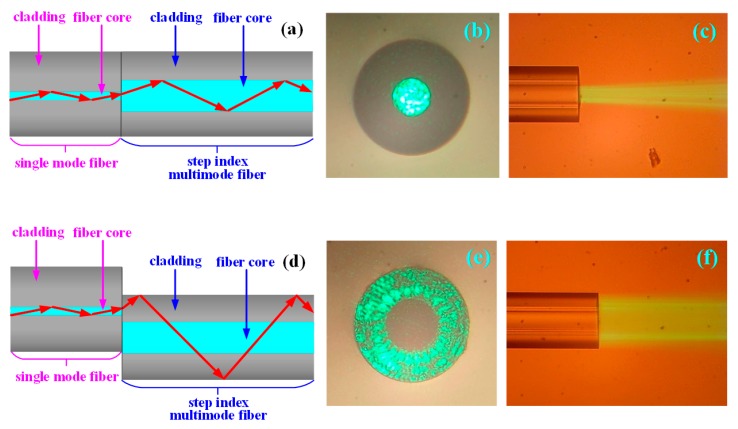
The investigation of the transmission of the light field of the step refractive index multimode fiber when normalized core welding and core-shift welding are used. (**a**) The schematic diagram of the structure and light transmission of the step refractive index multimode fiber when normalized welding is used. (**b**) Microphotograph of the light field from the end face of the step refractive index multimode fiber when normalized core welding is used. (**c**) Microphotograph of the light field from the side face of the step refractive index multimode fiber when normalized core welding issued. (**d**) The schematic diagram of the step refractive index multimode fiber structure and light transmission when core-shift welding is used. (**e**) Microphotograph of the light field from the end face of the step refractive index multimode fiber when core-shift welding is used. (**f**) Microphotograph of the light field from the side face of the step refractive index multimode fiber when core-shift welding is used.

**Figure 4 sensors-19-01202-f004:**
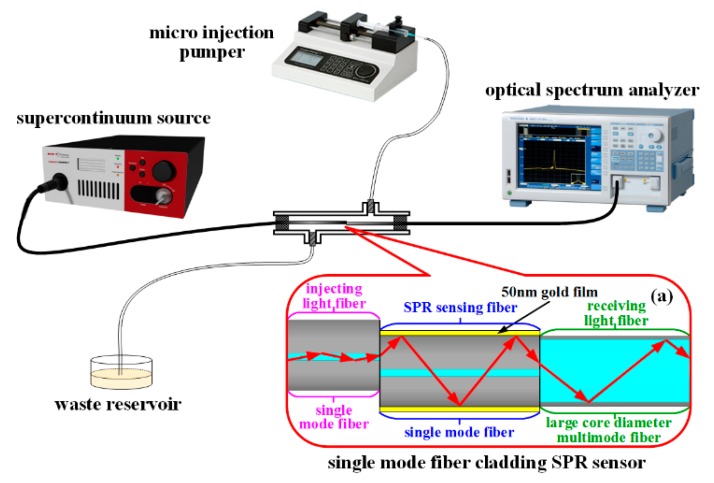
Diagram of the experimental test equipment of the single-mode fiber cladding SPR sensor based on the core-shift welding technology.

**Figure 5 sensors-19-01202-f005:**
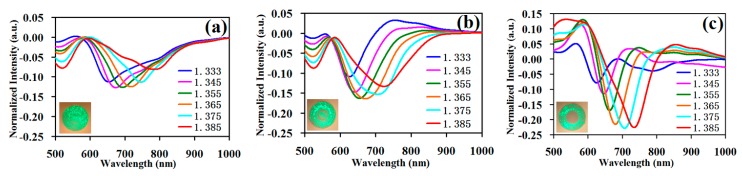
The testing data of all kinds of fiber cladding SPR sensors. (**a**) The testing data of the single-mode fiber cladding SPR sensor; (**b**) the testing data of the gradient refractive index multimode fiber cladding SPR sensor; (**c**) the testing data of the step refractive index multimode fiber cladding SPR sensor.

**Figure 6 sensors-19-01202-f006:**
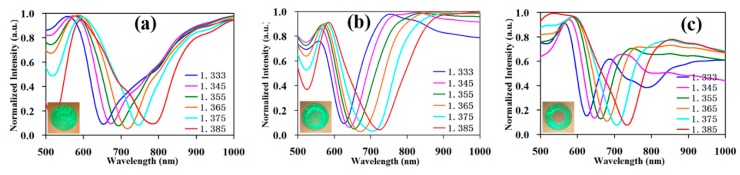
The testing data of the three types of fiber cladding SPR sensors are renormalized. (**a**) Normalization of the testing data of the single-mode fiber cladding SPR sensor; (**b**) normalization of the testing data of the gradient refractive index multimode fiber cladding SPR sensor; (**c**) normalization of the testing data of the step refractive index multimode fiber cladding SPR sensor.

**Figure 7 sensors-19-01202-f007:**
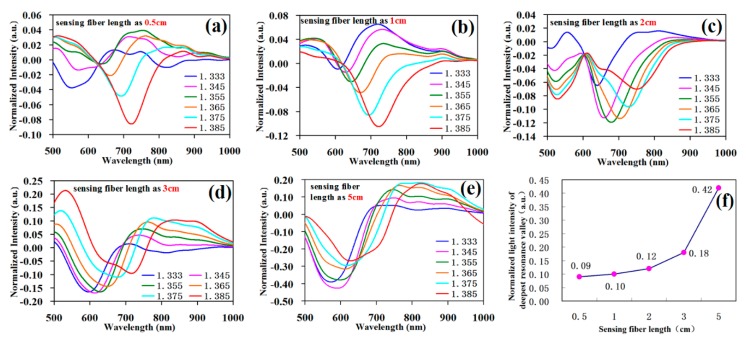
Testing data of the sensing fiber SPR with different lengths. (**a**) Experimental data of the sensing fiber length of 0.5 cm; (**b**) experimental data of the sensing fiber length of 1 cm; (**c**) experimental data of the sensing fiber length of 2 cm; (**d**) experimental data of the sensing fiber length of 3 cm; (**e**) experimental data of the sensing fiber length of 5 cm; (**f**) normalized light intensity of the deepest resonance valley corresponding to the sensing fibers of different lengths.

**Figure 8 sensors-19-01202-f008:**
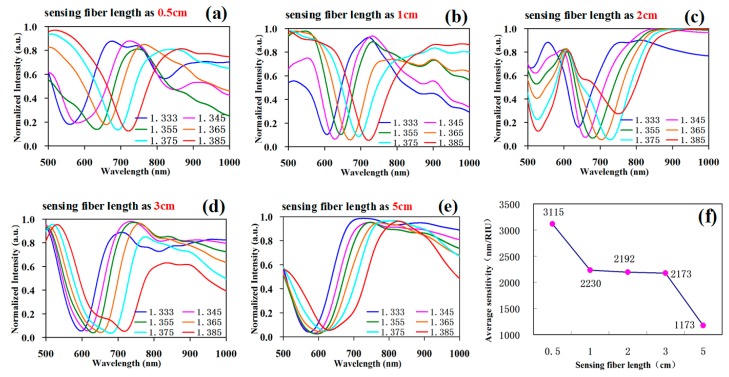
Renormalized sensing fiber SPR testing data with different lengths. (**a**) Normalized experimental data when the sensing fiber length was 0.5 cm; (**b**) normalized experimental data when the sensing fiber length was 1 cm; (**c**) normalized experimental data when the sensing fiber length was 2 cm; (**d**) normalized experimental data when the sensing fiber length was 3 cm; (**e**) normalized experimental data when the sensing fiber length was 5 cm; (**f**) corresponding relation diagram of sensing fiber length and sensitivity.

**Figure 9 sensors-19-01202-f009:**
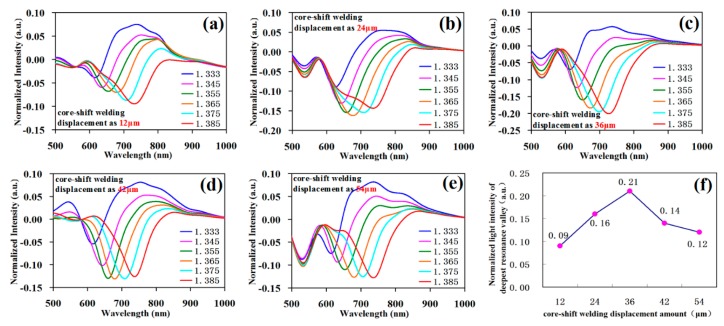
The SPR testing data with different displacements of the core-shift welding. (**a**) The experimental data of the core-shift welding displacement of 12 μm; (**b**) the experimental data of the core-shift welding displacement of 24 μm; (**c**) the experimental data of the core-shift welding displacement of 36 μm; (**d**) the experimental data of the core-shift welding displacement of 42 μm; (**e**) the experimental data of the core-shift welding displacement of 54 μm; (**f**) normalized light intensity of the deepest resonance valley corresponding to the core-shift welding displacement.

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
