# Peer review of "Optical Fiber Cladding SPR Sensor Based on Core-Shift Welding Technology"

_sensors, 2019, doi:10.3390/s19051202_

Reviewer 1 Report

Authors presented very interesting new approach for the reprodicible manufacturing method of optical fiber SPR sensors using core-shift welding. The manuscript is well organized but contains numerous mistakes and grammar issues and sometimes authors make strange choice of terms and verbs (e.g.  see the senstense "Based on the sensing mechanism, evanescent field must interact with 50nm gold film to produce SPR phenomenon and conduct sensing"). Such sentences are understadable, but definetely can be improved in order to be more "reader-friendly".

In addition, I suggest to remove excessive introductory part regarding the SPR sensing technique, as  most of the readers of this paper (as it has very specific topic) would have (at least) quite good understanding of SPR phenomenon.

Author Response

Response to reviewer 1:

Thank you very much for your letter and the comments from the referees about our paper. After carefully studying the comments and your advice, we have revised the paper according to your comments. The main revisions are listed as follows:

Question 1:

1. The manuscript is well organized but contains numerous mistakes and grammar issues and sometimes authors make strange choice of terms and verbs (e.g. see the senstense "Based on the sensing mechanism, evanescent field must interact with 50nm gold film to produce SPR phenomenon and conduct sensing"). Such sentences are understadable, but definetely can be improved in order to be more "reader-friendly".

Answer:

Thanks for the suggestion. We have checked and revised the grammer of the whole paper. Espescially, we revised the sentence “Based on the sensing mechanism, evanescent field must interact with 50nm gold film to produce SPR phenomenon and conduct sensing” into “Based on the sensing mechanism, SPR phenomenon is produced depending on the contact between evascent field of transmission light and gold field as the prerequisite to realize sensing”.

Question 2:

2. In addition, I suggest to remove excessive introductory part regarding the SPR sensing technique, as most of the readers of this paper (as it has very specific topic) would have (at least) quite good understanding of SPR phenomenon.

Answer:

Thanks for the suggestion. We have deleted the content on SPR detecting principle in the Introduction, according to your suggestion.

Reviewer 2 Report

Your paper entitled “Optical fiber cladding SPR sensor based on core-shift welding technology” is very interesting.  The gas detection and bio/chemical sensing community can benefit from an advanced surface plasmon resonance (SPR) sensor with increased sensitivities of detection.  Your approach to use a fiber cladding SPR sensor based on core-shift welding technology has advantages over existing methods such as cladding corrosion, side polishing, grinding on the end, optical gating, fused biconical taper, etc., which require complex processing for obtaining the interaction of evanescent field and the gold film.  Your unique technique solves the problem of hard acquisition of evanescent field in fiber type SPR sensors.

My comments/recommendations follow:

The phrase "kinds of fiber” is vague.  Recommend that “kinds of fiber” be replaced with “2 or 3 kinds of fiber”

The phrase “micro-structured fibers” is listed in the key terms, but never used in the transcript.  Recommend that the phrase “micro-structured fibers” be removed from the list of terms.

A sensitivity of 2538nm/RIU is first mentioned on page 8.  Recommend that this value be compared to other existing SPR sensor sensitivities.

In addition, an equation for the sensitivity, (Sλn), in terms of the shift in the resonance wavelength, ∂λres, and the unit change in the refractive index, ∂ns, would be useful in the paragraph below Fig. 6 on page 8 - (Sλn = ∂λres / ns).

Recommend that the fiber SPR sensing probe be identified in Fig. 4.

Recommend that the authors check the sentence structure through the manuscript.

I made a few suggestions to improve sentence structure (See the attached PDF document – Comments / Recommendations are in the yellow sticky notes).

Author Response

Response to reviewer 2:

Thank you very much for your letter and the comments from the referees about our paper. After carefully studying the comments and your advice, we have revised the paper according to your comments. The main revisions are listed as follows:

Question 1:

1. The phrase "kinds of fiber” is vague. Recommend that “kinds of fiber” be replaced with “2 or 3 kinds of fiber”

Answer: Thanks for the suggestion. In our manuscript, we have revised “kinds of fiber” into “three kinds of fiber”.

Question 2:

2. The phrase “micro-structured fibers” is listed in the key terms, but never used in the transcript. Recommend that the phrase “micro-structured fibers” be removed from the list of terms.

Answer: Thanks for the suggestion. In our manuscript, we have deleted “micro-structured fibers” from the list of terms.

Question 3:

3. A sensitivity of 2538nm/RIU is first mentioned on page 8. Recommend that this value be compared to other existing SPR sensor sensitivities.

Answer: Thanks for the suggestion. We have added the sentence ”The sensitivity of traditional fiber-core type SPR sensor with multi-mode fiber is about 2000nm/RIU.  For the novel fiber cladding SPR sensor we propose, it has a slightly higher sensitivity than most multimode fiber fiber-core type SPR sensor. ”

Question 4:

4. In addition, an equation for the sensitivity, (Sλn), in terms of the shift in the resonance wavelength, ∂λres, and the unit change in the refractive index, ∂ns, would be useful in the paragraph below Fig. 6 on page 8 - (Sλn = ∂λres / ∂ns).

Answer: Thanks for the suggestion. We have added the equation for the sensitivity in the paragraph below Fig. 6 on page 8 as the sentence “SPR equation for the sensitivity is Sλn = ∂λres / ∂ns, where Sλn is mean sensitivity, ∂λres is the shift in the resonance wavelength and ∂ns is the unit change in the refractive index”.

Question 5:

5. Recommend that the fiber SPR sensing probe be identified in Fig. 4.

Answer: Thanks for the suggestion. We have added the fiber SPR probe type of single mode fiber cladding SPR sensor in Figure 4.

Question 6:

6. Recommend that the authors check the sentence structure through the manuscript. I made a few suggestions to improve sentence structure (See the attached PDF document – Comments / Recommendations are in the yellow sticky notes).

Answer: Thanks for the suggestion. We have checked and revised the grammer of the whole paper.